# Mitochondrion-Targeted NIR Therapeutic Agent Suppresses Melanoma by Inducing Apoptosis and Cell Cycle Arrest via E2F/Cyclin/CDK Pathway

**DOI:** 10.3390/ph15121589

**Published:** 2022-12-19

**Authors:** Changzhen Sun, Jianv Wang, Tong Xia, Qin Sun, Yijing He, Hailan Wang, Qizhou He, Li Liu

**Affiliations:** 1Drug Research Center of Integrated Traditional Chinese and Western Medicine, National Traditional Chinese Medicine Clinical Research Base, The Affiliated Traditional Chinese Medicine Hospital of Southwest Medical University, Luzhou 646610, China; 2Department of Dermatology, The Affiliated Hospital of Southwest Medical University, Luzhou 646000, China; 3Department of Science and Technology, The Affiliated Hospital of Southwest Medical University, Luzhou 646000, China; 4Department of Radiology, The Affiliated Traditional Chinese Medicine Hospital of Southwest Medical University, Luzhou 646610, China

**Keywords:** near-infrared imaging, heptamethine cyanine dyes, malignant melanoma, apoptosis, cell cycle arrest

## Abstract

Malignant melanoma is the most fatal form of skin cancer worldwide, and earlier diagnosis and more effective therapies are required to improve prognosis. As a possible solution, near-infrared fluorescent heptamethine cyanine dyes have been shown to be useful for tumor diagnosis and treatment. Here, we synthesized a novel theranostic agent, IR-817, a multifunctional bioactive small-molecule that has near-infrared emission, targets mitochondria in cancer cells, and has selective anti-cancer effects. In in vitro experiments, IR-817 preferentially accumulated in melanoma cells through organic anion transporting polypeptide transporters but also selectively inhibited the growth of tumor cells by inducing mitochondrial-dependent intrinsic apoptosis. Mechanistically, IR-817 caused G0/G1 cell cycle arrest by targeting the E2F/Cyclin/CDK pathway. Finally, IR-817 significantly suppressed the growth of xenograft tumors in zebrafish and mice. Immunohistochemical staining and hematoxylin and eosin staining revealed that IR-817 induced apoptosis and inhibited tumor cell proliferation without notable side effects. Therefore, mitochondrial-targeting theranostic agent IR-817 may be promising for accurate tumor diagnosis, real-time monitoring, and safe anti-cancer treatments.

## 1. Introduction

Malignant melanoma (MM) is the rarest but most serious form of skin cancer. Although it makes up only 5–10% of all skin cancers, MM accounts for more than 75% of skin cancer-related deaths [1,2]. A worldwide total of 325,000 new melanoma cases and 57,000 deaths was estimated for 2020. If 2020 rates continue, the burden from melanoma is estimated to be a roughly 50% increase and a 68% increase in deaths by 2040 [3]. Unlike other solid tumors, MM usually occurs in patients younger than 60 years. The early stages of MM are relatively easy to treat using surgical intervention. However, because melanoma is highly invasive and metastatic, the prognosis for long-term survival is poor. The five-year relative survival rate for patients with stage 0 melanoma is 97%, compared to stage IV disease, with approximately 10% survival. Although targeted therapies and immunotherapies have improved clinical outcomes, melanoma remains difficult to treat once it metastasizes to other sites, which include the brain, lungs, liver, or bone [4]. Therefore, image-guided, multimodal cancer theranostic strategies are needed for early diagnosis and improved melanoma treatments.

Mitochondria, the powerhouse of the cell, are organelles that maintain normal cellular function and metabolism, which are potential targets for therapeutic intervention all this time [5]. Mitochondrial dysfunction is involved in many human diseases, such as cancer, metabolic disorders, and cardiovascular and neurodegenerative diseases [6]. Therefore, mitochondria are promising targets for both diagnosis and treatment, and they have garnered increased interest in medical and pharmaceutical research [7]. Mitochondria-targeted therapeutic strategies are considered promising approaches for cancer therapy [8]. However, owing to their unique lipid bilayer structure and negative potential, therapeutic molecules have difficulty reaching the mitochondrial interior [9]. At present, some mitochondria-specific agents have been developed by covalent conjugation with mitochondria-targeting units, such as triphenylphosphonium, rhodamine derivatives, cyanine dyes, and mitochondria-targeting peptides [10]. Cyanine dyes exhibit excellent mitochondria-targeting capacities owing to their lipophilic and cationic properties. These properties make cyanine dyes attractive conjugates for anti-melanoma treatment by inducing mitochondria-targeted apoptosis.

Fluorescence imaging agents in the near-infrared (NIR), within the wavelength range of 700–900 nm, have attracted widespread attention for both non-invasive tumor imaging and therapy [11,12]. Compared with visible wavelengths, imaging within the NIR region has the advantages of deep tissue penetration, low background autofluorescence, and low tissue absorbance and scattering [13,14,15]. Hence, novel NIR agents are favorable for image-guided anti-cancer therapy. However, theranostic agents having NIR fluorescence, tumor targeting, and treatment capabilities are limited [15]. The combination of NIR fluorophores with tumor-targeting ligands and/or anti-cancer drugs is a broadly applicable strategy for image-guided anti-cancer therapy [12,16,17]. Although significant progress has been made in cancer theranostics, there is still uncertainty regarding the delivery, targeting, and therapeutic potential of this strategy because chemical conjugation may alter ligand tissue targeting [18]. Therefore, there is an urgent need to develop small-molecule NIR fluorophores for both cancer-targeted imaging and therapeutics. In a previous study, the multifunctional small-molecule fluorescent probe, ZWZ-3, was used for melanoma imaging and therapy [19]. ZWZ-3 preferentially accumulates in the mitochondria of tumor cells using the organic anion-transporting polypeptide (OATP) transporters. ZWZ-3 induces apoptosis through the reactive oxygen species mitochondrial apoptotic pathway and promotes autophagy.

In this study, we synthesized and characterized melanoma-targeted IR-817, a small-molecule NIR fluorescent heptamethine cyanine dye that accumulated in cancer cell mitochondria for tumor imaging. Furthermore, mechanistic analysis revealed that IR-817 treatment induced apoptosis through the mitochondrial apoptosis pathway and induced G0/G1 cell cycle arrest by targeting the E2F/Cyclin/CDK pathway. Our data provide a practical strategy for developing heptamethine-based cyanine NIR fluorescent dye theranostic agents for simultaneous cancer imaging, targeting, and therapy (Figure 1).

## 2. Results

### 2.1. Chemical Synthesis and Optical Properties of IR-817

A schematic representation of the IR-817 synthetic route and the structural formula of IR-817 are shown in Appendix A and Figure 1A. IR-817 was obtained by coupling NIR probe 808 with choline via a redox reaction. The structure and conformation of IR-817 were confirmed by HRMS, ^1^H NMR, and ^13^C NMR (Appendix A). Subsequently, IR-817 was characterized by spectroscopic analysis. The absorption and fluorescence spectra of IR-817 were recorded in MeOH, 10% FBS, and H_2_O. As shown in Figure 1B,C, IR-817 exhibited a maximum absorption peak (764 nm) as well as an emission peak (790–820 nm) in the NIR region (700–900 nm). To investigate the correlation between fluorescence intensity and concentration, serially diluted IR-817 was tested in different solvents, namely MeOH, 10% FBS, and H_2_O. The fluorescence intensity of IR-817 correlated with concentration, particularly in H_2_O, with a high linear correlation index (R^2^ = 0.9932) (Figure 1D,E and Appendix A). Although IR-817 had a strong fluorescence intensity in MeOH, the fluorescence intensity decreased after 20 min (Appendix A). This indicates that the fluorescence signal of IR-817 in MeOH was unstable, and MeOH was not an appropriate solvent for IR-817. In contrast, the fluorescence intensity in 10% FBS was stable and reached a plateau within 120–720 min (Figure 1F and Appendix A), which demonstrated that IR-817 is stable in aqueous environments and is quite can be used for live cell imaging.

### 2.2. Identification of IR-817 for Melanoma Targeting Imaging In Vitro and In Vivo

To determine any targeting effect of IR-817 in vitro, four cancer cell lines (A549, HeLa, A375, and B16-F10) and one normal cell line (COS-1) were incubated with IR-817. Representative results are shown in Figure 2A,B and Appendix A. IR-817 preferentially accumulated in cancer cells compared to COS-1, especially in melanoma cell lines A375 and B16-F10. The fluorescence of IR-817 in B16-F10 was weaker than that in A375, possibly because the B16-F10 cells contained melanin. We also demonstrated this effect in mice. In vivo, NIR fluorescence imaging of B16-F10-bearing tumor xenografts revealed that IR-817 preferentially accumulated in tumor tissues (Figure 2C). Red fluorescence from IR-817 was observed in the tumor but not in other tissues except the liver (Appendix A). We further demonstrated IR-817 selectivity for tumor tissues by the lack of fluorescence from dissected organs. In addition, in vivo imaging was performed for 48 h, and the tumor fluorescence intensity was reduced but visible, yet fluorescence was augmented in the liver (Appendix A). These results indicated that IR-817 could selectively target the tumor in vivo, had good stability with a residence time of up to 48 h, and could mostly be metabolized through the liver. The latter result is important to demonstrate that IR-817 could be degraded by first-pass metabolism rather than damaging other tissues.

### 2.3. IR-817 Specifically Targeted Mitochondria and Entered Cells through OATP Transporters

We investigated the tumor-targeting mechanism of IR-817 using in vitro tests. Because the heptamethine cyanine dye is a lipophilic cationic structure that can preferentially accumulate in the mitochondria, we investigated whether IR-817 targeted the mitochondria [20,21,22]. To determine whether IR-817 can be largely located in mitochondria, IR-817 was added before co-incubation with Mito-Tracker Green and Lyso-Tracker Green fluorescent dyes. IR-817’s red fluorescence highly overlapped with the mitochondrial green fluorescence, whereas it had a low degree of overlap with lysosomal green fluorescence in A375 and B16-F10 cells (Figure 2D and Appendix A). Merged areas appeared in the mitochondria rather than lysosomes, which was highly correlated in both A375 and B16-F10 cells, and Pearson’s coefficient was 0.911 and 0.920, respectively (Figure 2E and Appendix A). Moreover, melanoma cells were pretreated with BSP, a competitive inhibitor of OATP transporters, and the cellular uptake of IR-817 was remarkably reduced (Figure 2F,G), indicating that IR-817 enters into the cells through OATP transporters and specifically accumulates in the mitochondria, as has been indicated for other previously reported cancer-targeted heptamethine cyanine dyes [14,23]. 

### 2.4. IR-817 Induced Apoptosis of Melanoma Cells A375 and B16-F10 through the Mitochondrial Apoptotic Pathway

Flow cytometry (FCM) was performed in A375 and B16-F10 cells to explore whether IR-817 can induce melanoma cell apoptosis. As shown in Figure 3A, increasing IR-817 concentrations correlated with increased levels of late apoptotic cells. The late apoptosis levels of A375 and B16-F10 cells treated with 2.5 μM IR-817 rose more than 50% compared to untreated cells (Figure 3B). We also visualized the apoptotic process using Hoechst 33342 staining. After treatment with IR-817, the number of apoptotic cells increased, accompanied by apoptotic body production in a concentration-dependent manner (Appendix A). These data indicated that IR-817 induced melanoma cell apoptosis in a dose-dependent manner.

We have previously established that IR-817 localizes to the mitochondria. Mitochondria play an important role in the intrinsic apoptotic pathway [24]. Loss of mitochondrial membrane potential (ΔΨm) caused by membrane permeability disruption is an important event in this apoptotic pathway [25]. We examined the loss of ΔΨm by FCM using JC-1 staining to evaluate whether apoptosis was caused by mitochondrial membrane permeability disruption. When the membrane potential decreased, JC-1 showed a monomer state and glowed green. On the contrary, normal membrane potential caused JC-1 to enter the cell and aggregate into the polymer, emitting red light [26,27]. We found that IR-817 had a dose-dependent effect in disrupting the ΔΨm in both A375 and B16-F10 cells, compared with the untreated cells, the monomeric ratio of JC-1 was from 7.94% to 39.3% and 31.66% to 87.12% in A375 and B16-F10 cells, respectively (Figure 3C). The decreasing membrane potential was more clearly shown by the green/red fluorescence ratio (Figure 3D). These data suggest that IR-817 induces cell apoptosis by reducing ΔΨm and thus promoting the mitochondrial apoptotic pathway. Subsequently, western blotting was performed to determine the effects of IR-817 on Bcl-2 family proteins. Bcl-2 levels decreased, and Bax levels increased with increasing concentrations of IR-817 in B16-F10 cells, resulting in an increased ratio of Bax to Bcl-2 (Figure 3E and Appendix A). In addition, the expression levels of cleaved caspase-3 (CC-3) and cleaved caspase-9 (CC-9) proteins were measured and compared with the untreated group, their expression was significantly increased. In conclusion, IR-817 induced mitochondrial-induced apoptosis in melanoma cells and was associated with the activation of the caspase cascade.

### 2.5. IR-817 Selectively Inhibited Melanoma Cell Proliferation

The effectiveness of IR-817’s anti-tumor proliferation capability was evaluated using the MTT assay. A375, B16-F10, A549, MDA-MB-231, and HeLa cell viability was measured after co-incubation with different concentrations of IR-817 for 72 h. IR-817 reduced the proliferation of the melanoma cell lines (A375, B16-F10, and A549) and was less effective against other cancer cell types (MDA-MB-231 and HeLa) (Figure 4A). This result indicated that these melanoma cell lines were more sensitive to IR-817. In addition, IR-817 suppressed melanoma cell proliferation in a time- and dose-dependent manner (Figure 4B,C) but had no inhibitory effects on normal cells (Figure 4D). The direct observation of cell morphological changes under the microscope is also an intuitive and simple way to show the effects of drugs [28]. Therefore, we observed any structural changes in these cells with optical microscopy. A375 and B16-F10 cell numbers decreased with increasing IR-817 concentrations compared to the control group. With higher doses, these cells detached and became small and rounded (Appendix A). Furthermore, colony formation was evaluated to investigate whether IR-817 reduced melanoma cell line proliferation. The number of colonies decreased after treatment with different concentrations of IR-817 (Figure 4E). Moreover, the colony size in the IR-817 treatment group was notably smaller than that in the control group. These results were consistent with the MTT data and suggested that IR-817 inhibited melanoma cell proliferation in a concentration- and time-dependent manner.

### 2.6. IR-817 Induced Melanoma Cell Cycle Arrest at the G0/G1 Phase by E2F/Cyclin/CDK Signal Regulation Network

To determine the mechanism by which IR-817 inhibits proliferation and promotes apoptosis of melanoma cells, RNA sequencing of B16-F10 cells was performed after IR-817 treatment. We further performed differential expression analysis between the two groups. A total of 1361 DEGs were identified, including 770 upregulated and 591 downregulated genes in the high mRNA group (Figure 5A). Then, functional enrichment analysis was performed with the DEGs, and DNA damage repair pathway-related genes were significantly down-regulated following IR-817 treatment (Figure 5B). We also constructed a protein-protein interaction network using STRING. For the 72 genes identified in the DNA repair pathway, a total of three significant modules were identified (Appendix A). Among the three modules, 26 proteins in the yellow module were mainly cyclin-related proteins. Therefore, we assessed IR-817’s effect on cell cycle progression using flow cytometry. As shown in Figure 5C, increasing IR-817 doses correlated with increased ratios of G0/G1 phase arrest in both A375 and B16-F10 cells, as the ratio of G0/G1 phase arrest in A375 cells was from 53.62% to 68.57% and in B16-F10 cells was from 48.72% to 71.23%. These data showed that IR-817 induced cell cycle arrest at the G0/G1 phase, contributing to the suppression of cell proliferation.

Based on the gene clustering and cell cycle arrest results, we used qRT-PCR to assess transcript levels of genes (E2f8, E2f7, Cdc45, Ccna2, Cdk1, Ccne1, and Rrm2) that are cyclin-related and have been reported to block the G0/G1 phase [29,30]. Excepting E2f7, all relative differences were consistent with the sequencing results, and E2f8 was significantly downregulated (Figure 6A). E2f8 was selected as the key gene because it was highly downregulated after IR-817 treatment and was a core protein in the cyclin-related PPI network (Figure 6B). Other contributing factors are that E2f8 is involved in transcription, and its gene copy number is increased in melanoma [31]. This result indicates that IR-817 may be highly applicable for clinical melanoma treatment, but more studies are required.

In order to clarify the effect of IR-817 on the cell cycle, we selected Cyclin D1, Cyclin E1, Cdk2, Cdk4, P21, and P27 as the target proteins for verifying cell cycle changes, which were the key proteins for regulating the cell cycle. As the results show, the expression of E2f8, Cyclin D1, Cyclin E1, Cdk2, and Cdk4 was downregulated after treatment with IR-817 (Figure 6C–F and Appendix A). IR-817 induced the accumulation of P21 and P27, which are two vital intracellular cell cycle inhibitors. These experimental results confirmed that IR-817 could both target E2f8 and induce cell cycle arrest at the G0/G1 phase through the E2F/Cyclin/CDK signaling pathway, thereby contributing to the suppression of cell proliferation.

### 2.7. IR-817 Inhibited the Growth of Melanoma In Vivo

In recent years, zebrafish xenotransplantation has been demonstrated to be an accurate model of human cancer and can be rapidly used for in vivo assessment of anti-cancer drugs and drug targets owing to its unique advantages. Additionally, zebrafish have emerged as a valuable model system for studying melanoma [32,33]. The significant anti-tumor effect of IR-817 in vitro motivated us to further investigate its therapeutic efficiency in vivo. In zebrafish, the melanoma cell lines GFP-A375 and Dil-B16-F10 were used to evaluate the anti-tumor effects of IR-817. Zebrafish inoculated with either GFP-A375 and Dil-B16-F10 cells were cultured in a medium that contained IR-817, and any alterations in fluorescence were recorded. IR-817 significantly reduced the fluorescence intensity and size of tumor cells reduced compared to the control group (Figure 7A,B). Therefore, IR-817 inhibited the growth of tumor cells in a zebrafish tumor model in a concentration-dependent manner.

We have identified that IR-817 has good anti-tumor ability in vitro and in zebrafish. To further investigate the in vivo effect, we used the B16-F10 tumor-bearing mouse model treated every other day with IR-817 (2 mg/kg). The tumors in IR-817-treated mice were smaller than those in the control group (Figure 8A). Compared with the control group, IR-817 significantly inhibited the volume and weight of tumor tissue without causing significant body weight loss (Figure 8B–D). The tumor growth inhibition rates on day 15 post-inoculation were greater than 50%. Moreover, further investigations revealed that changes in body weight and H&E organ staining showed that IR-817 did not alter body weight, cell morphology, or tissue architecture of mice (Figure 8E,F), which indicated that IR-817 might have had high biological safety and almost no toxic side effects in this study. Consistent with the In vitro data, IHC staining analysis of the B16-F10 mouse model tumors showed that IR-817 suppressed E2f8 expression and the proliferation marker Ki-67 (Figure 8G). Furthermore, considerable CC-3 staining intensity showed IR-817-induced apoptosis in tumor tissues. Collectively, these data suggest that IR-817 has potent anti-melanoma effects in vivo.

## 3. Discussion

Because of melanoma’s high invasiveness and metastatic ability, which needs early diagnosis and surgical resection to get satisfactory treatment effects; however, many patients cannot be diagnosed in time [4]. Therefore, image-guided, multimodal cancer treatment strategies are necessary for early diagnosis and improved treatment of melanoma. Despite the fact that there have been a lot of related reports on heptacyanine dyes in cancer diagnosis and treatment, due to the different ways chemical coupling will change the corresponding targeting and therapeutic effect, it is still necessary to continue in-depth research and develop NIR fluorescent small molecule compounds with better targeting and therapeutic potential. In this study, we obtained a multifunctional bioactive small molecule IR-817 by the one-step redox reaction of IR-808 with choline, which not only has near-infrared emission but also targets mitochondria in cancer cells and can selectively inhibit melanoma.

Mitochondrial-mediated apoptosis is tightly regulated by Bcl-2 family proteins, including pro-apoptotic Bax and anti-apoptotic Bcl-2 [34]. They play a pivotal role in the mitochondrial apoptotic pathway by regulating the outer membrane’s permeability, and increased permeability can lead to the release of cytochrome c, caspase 3/9 activation, and cell death [35,36,37]. Our study confirmed that IR-817 could promote Bax and inhibit the expression of Bcl-2 protein at the same time, thus leading to the increase of the ratio of Bax to Bcl-2 and further triggering the caspase cascade reaction. This is the intrinsic mechanism of IR-817 promoting apoptosis of melanoma cells.

By RNA sequencing, GO pathway enrichment, and PPI-related gene analysis methods, E2f8 was identified as the target of the IR-817 regulatory cell cycle. As transcriptional regulators, the E2F family plays a crucial role in regulating cell functions, including DNA damage repair and the cell cycle [38]. E2f8 has been reported in liver cancer, lung cancer, cervical cancer, and other related diseases as it represses E2F transcriptional activation. However, there are few reports of E2f8 activity in melanoma [39]. An abnormal cell cycle is a hallmark of cancer, and many proteins are involved in cell cycle regulation. It has been reported that E2f8 negatively affects cell cycle progression by interacting with several cycle-related proteins, promotes the G1 to S phase transition, regulates transcriptional activity, and promotes cell proliferation [40,41]. During the cell cycle, Cyclin/CDK/CKI signal regulation network existed, and Cyclin positively regulated cyclin-dependent kinase (CDK) as well, as CDK inhibitor (CKI) negatively regulated CDK [42,43,44]. P27 is a classic CKI that interacts with Cyclins, Cdk2, and Cdk4, to regulate the cell cycle at the G1/S transition [45]. We confirmed that IR-817 could target E2f8 and induce cell cycle arrest at G0/G1 phase through E2F/Cyclin/CDK signaling pathway, thereby inhibiting cell proliferation.

## 4. Materials and Methods

### 4.1. Reagents and Antibodies

IR-808 was synthesized according to the reported method [46,47,48]. Dicyclohexylcarbodiimide, N, N-dimethylformamide, dichloromethane, methanol, sodium chloride, and dichloromethane were purchased from KeLong Chemicals (Chengdu, China). Dulbecco’s modified Eagle’s medium (DMEM) and fetal bovine serum (FBS) were purchased from BasalMedia (Shanghai, China), trypsin-EDTA, penicillin-streptomycin, cisplatin, PMSF, and crystal violet staining solution were supplied by Solarbio (Shanghai, China). 3-(4,5-diMethylthialzol-2-yl)-2,5-diphe-nyltetrazolium bromide (MTT) and Hoechst33342 were purchased from YuanYe Bio-Technology (Shanghai, China). AnnexinV-FITC/PI was purchased from Biosharp (Hefei, China). JC-1 MMP assay kit and ECL detection were purchased from Yeasen (Shanghai, China). MitoTracker Green was purchased from Beyotime, China. Sulfobromophthalein (BSP) was purchased from Alfa Aesar (Shanghai, China). The universal SP kit (mouse/rabbit streptavidin–biotin detection system) and DAB kit were purchased from Zsbio (Beijing, China). Dil cell membrane dye (Molecular, probes), RIPA lysate buffer, BCA protein determination kit, and antigenic repair solution was from Biyuntian Biotechnology (Shanghai, China). The protein loading buffer was from Biotechnology Co., Ltd. (Beijing Zoman). 8–12% SDS-PAGE and the rapid blocking solution were from EpiZyme (Shanghai, China). RNA-Easy Isolation Reagent, RNA-Easy Isolation Reagent, and SYBR Premix Ex Taq kit were purchased from Invitrogen Technologies Company (Carlsbad, CA, USA).

Antibodies against E2f8 were purchased from SAB ((Baltimore, MD, USA)). Antibodies against Bax were purchased from Cell Signaling Technology (CST, Beverly, MA, USA). Antibodies against Cleaved caspase-9 were purchased from Abcam (Abcam, Cambridge, UK). Antibodies against Cleaved caspase-3, β-actin, GAPDH, and anti-mouse HRP, anti-rabbit HRP were purchased from Zen Bioscience were purchased from Zen-bio (Chengdu, China). Antibodies against Bcl-2, Cdk2, Cdk4, Cyclin D1, Cyclin E1, P21, and P27 were purchased from the Beyotime Institute of Biotechnology (Shanghai, China).

### 4.2. Synthesis of IR-817

IR-808 (764 mg, 1 mm) and dicyclohexylcarbodiimide (824 mg, 4 mm) was dissolved in N, N-dimethylformamide (20 mL) and stirred at room temperature for 1 d. A dichloromethane/methanol ratio of 10:1 was used as a solvent and detected by TLC. After the reaction was completed, it was quenched with a saturated aqueous solution of sodium chloride, washed twice, and extracted with dichloromethane. The organic phase was concentrated by rotary evaporation under reduced pressure and purified by a silica gel column using dichloromethane/methanol as the elution solvent. The target product was collected after concentration under reduced pressure (0.95 g, 81%). Cal. M^+^: C_68_H_96_ClN_6_O_4_^+^, 1095.7176, found: 1095.7130. The product was characterized by ^1^H NMR (CD_3_OD-d_4_, 400 MHz): δ = 0.92–1.48 (40H, m), 1.45–1.90 (26H, m), 2.45–2.55 (2H, m), 2.70–2.80 (2H, m), 3.20–3.40 (4H, m), 3.70–3.95 (2H, m), 4.35–4.50 (2H, m), 5.50–5.60 (2H, m), 6.75–6.85 (2H, m), 7.25–7.32 (1H, t, J = 8), 7.40–7.48 (2H, m), 7.60–7.65 (1H, d, J = 7.4), 8.20–8.40 (2H, m).

^13^C NMR (DMSO-d6, 101 MHz): δ 172.69, 169.81, 154.01, 143.44, 142.50, 141.51, 126.65, 112.04, 102.14, 52.96, 49.94, 49.45, 34.23, 33.76, 32.14, 30.77, 27.91, 26.11, 25.90, 25.54, 25.12, 24.83.

The preparation of IR-817 solutions: 2.8 mg IR-817 was dissolved in 119 μL DMSO to obtain 20 mM IR-817 storage solution, which was stored at 4 °C away from light.

### 4.3. Cell Culture

A375, B16-F10, MDA-MB-231, A549, HeLa, HEK-293T, and COS-1 were obtained from ATCC (Manassas, VA, USA), GFP-A375 cells were given away by Dr. F. Jianguo (Southwest Medical University, Luzhou, China). and were cultured in Dulbecco’s modified Eagle’s medium (DMEM) containing 10% fetal bovine serum and 1% penicillin/streptomycin. All the cells were cultured in an atmosphere of 5% CO_2_ at 37 °C, and the passage times were no more than five times.

### 4.4. Animal Models

C57BL/6J female mice between four to six weeks old were purchased from the Chengdu Dossy Laboratory Animals Company (Chengdu, China) and housed under conventional conditions. 100 μL B16-F10 cells (1–5 × 10^6^ cells/mL) were intradermally injected in haunch to establish a subcutaneous tumor model. All animal experiments have been approved by the Laboratory Animal Management Committee of the Affiliated Hospital of Southwest Medical University in China (Permit Number: 201903-39) and were conducted in accordance with the approved guidelines.

### 4.5. Fluorescence Optical Properties

To detect the optical properties of IR-817 in different solvents, IR-817 was dissolved in methanol or 10% FBS or H_2_O to obtain different concentrations at room temperature. The absorbance was measured on a UV-Vis scanning spectrophotometer Puxi TU-1900, and fluorescence intensities were detected using an FS5 spectrofluorometer (Edinburgh Instruments, Livingston, UK) with an excitation wavelength of 764 nm.

### 4.6. Intracellular Imaging

In order to study the selective imaging effect of IR-817, A375, B16-F10, Hela, A549, and COS-1, cells were seeded in 35 mm cell glass-bottom dishes with a density of 1 × 10^5^ cells/well. After 5 μM IR-817 co-incubated 1 h, the fluorescence images were captured using a confocal laser-scanning microscope (FluoViewTM FV1000 confocal microscope, Olympus Imaging) with 640 nm excitation and 778 nm emission.

### 4.7. Fluorescence Imaging In Vivo

C57BL/6J mice bearing intradermal tumors were injected via the tail vein with 0.2 mL IR-817 at a dose of 5 mg/kg (n = 5). After injection for 12 h and 48 h, the hair on the backs of the mice was shaved, and nebulized inhalation of methoxhalothane was used for anesthesia. Then, mice were imaged using the IVIS Lumina Series III Imaging System (PerkinElmer, Baltimore, MD, USA) equipped with fluorescent filter sets (excitation/emission, 620/710 nm).

### 4.8. Imaging of Subcellular Localization

A375, B16-F10 cells were seeded in 35 mm glass-bottom cell culture dishes and treated with 5 μM IR-817 for 1 h, then stained with Lyso-Tracker Green for 25 min or Mito-Tracker Green for 45 min. The fluorescent images were recorded using a confocal laser-scanning microscope (Olympus Fluo View FV1000) with 640 nm excitation and 778 nm emission.

### 4.9. The Cellular Uptake Mechanism of IR-817

The cellular uptake mechanism of IR-817 was studied by the BSP, an inhibitor of organic anion transport proteins (OATP) [49]. BSP (250 µm) was added to the six-well plates that were seeded with A375, B16-F10 cells for 20 min and incubation with 5 µm IR-817 for 1 h. Then, 1 μg/mL Hoechst33342 was added to the six-well plates for another 10 min after washing with PBS. The imaging was performed on an inverted fluorescence microscope (Leica DMi8, Leica, Germany).

### 4.10. Cells Apoptosis Assay

Cell apoptosis was detected according to Annexin V-FITC/PI detection kit and Hoechst33342. For Annexin V-FITC/PI assay, A375 and B16-F10 cells were collected after IR-817 (0, 0.31, 0.63, 1.25, 2.5 µM) incubated for 24 h, 5 μL Annexin V-FITC was co-incubated at room temperature in the dark for 20 min and 10 μL PI was added before flow cytometry (BD Biosciences, USA). Data analysis was carried out using FlowJo software.

Hoechst 33342 is a permeable fluorescent dye that makes the rounded nuclei of normal cells light blue and the nuclei of apoptotic cells dark blue and fragmented forms [50]. After A375 and B16-F10 cells respectively incubated with IR-817 (0, 0.31, 0.63, 1.25, 2.5 µM) for 24 h, 10 μg/mL Hoechst33342 were added for 30 min after the cells cleaning with PBS. Then observing and photographing under a fluorescence microscope (Leica DMi8, Leica, Germany).

### 4.11. Mitochondrial Membrane Potential (ΔΨm) Detection Assay

According to the instructions of the JC-1 mitochondrial membrane potential detection kit, A375, and B16-F10 cells were stained with JC-1 at 37 °C for 20 min, then washed and suspended with JC-1 buffer solution, and finally measured by FACS Calibur flow cytometry (Novocyte, ACEA Bio, San Diego, CA, USA) for changes in red and green fluorescence intensity. Excitation wavelength Ex = 490 nm and emission wavelength FL1 (Em = 530 nm), FL2 (Em = 590 nm).

### 4.12. Western Blotting

An application of standard western blots was performed to determine the level of protein expression. The cell total protein was extracted with RIPA lysate buffer containing 10% PMSF and collected the supernatant after the ultrasonication step. The extracted protein was measured with a BCA protein determination kit and boiled with a protein loading buffer. The samples were separated by 8–12% SDS-PAGE and transferred to the PVDF membrane. The transformed PVDF membrane was blocked in rapid blocking solution at room temperature for 15 min and then immersed in objective primary antibody diluent. After being incubated overnight at 4 °C, the membranes were incubated with enzyme-conjugated IgG buffer (anti-rabbit or anti-mouse) for 1.5 h at room temperature. The proteins were exposed by ECL reagents and visualized using UVP ChemStudio PLUS (Analytikjena) and were quantified by Image J software.

### 4.13. Anti-Proliferation Assay In Vitro

For the MTT assay, cells were seeded in a 96-well plate overnight and incubated in different concentrations of IR-817 (0.08, 0.16, 0.31, 0.63, 1.25, 2.5, 5.0, 10.0 µm) for 24, 48, and 72 h respectively. After reaching the scheduled time, cells were incubated with 20 μL MTT (the stock concentration of MTT was 5 mg/mL) for 2 h at 37 °C. Lastly, we discarded the liquid and incubated 100 µL DMSO for 10 min, the OD value at 570 nm was detected using the microplate tester (Spectra MAX 340, Molecular Devices, San Jose, CA, USA).

For the colony formation assay, A375, B16-F10 cells were seeded in six-well plates overnight. Then, the different concentrations of IR-817 (0, 0.31, 0.63, 1.25, 2.5 µm) were added into the medium and incubated for 7 days. The medium containing IR-817 was changed every three days. Finally, after washing with PBS, the cells were fixed with 4% formaldehyde and stained with 0.5% crystal violet solution, photographs were taken and counted.

### 4.14. RNA Sequence and Analysis

B16-F10 cells treated with or without 1.25 µm IR-817 for 24 h, sequencing analysis was performed by BGI (Shenzhen, China). The bioinformatics analysis methods of RNA sequence genes are as follows. The heatmap was drawn by pheatmap (v1.0.12) according to the gene expression in different samples. Differentially expressed genes (DEGs) were identified using the DESeq2 package (v1.4.5) under the criterion of the adjusted *p* < 0.05 and |log2FC| > 1. Kyoto Encyclopedia of Genes and Genomes (KEGG) is an encyclopedia for understanding biological functions and related genetic or genomic information. The KEGG database provides a series of pathway maps showing various known metabolic pathways and regulatory pathways, which can be used to analyze the related roles and functions of target genes [51]. In order to explore the main functions and roles of these differential genes, The cluster Profiler Bioconductor packages were used to enrich gene ontologies (GO) and the Kyoto Encyclopedia of Genes and Genomes (KEGG) (v4.2.2) (http://www.bioconductor.org/, accessed data: 22 March 2022), and the adjustment *p*-value < 0.05 was considered to be a functional pathway. STRING is a way to analyze protein-protein interactions (PPI) by collecting and integrating known or predicted association data between proteins from a variety of organisms [52]. Sequence data were submitted to the NCBI Sequence Read Archive under BioProject ID PRJNA866290 and PRJNA882993.

### 4.15. Quantitative Real-Time Reverse Transcript PCR (qRT-PCR)

Total RNA was extracted from B16-F10 cells using RNA-Easy Isolation Reagent according to the manufacturer’s instructions. The RNA was further reverse transcribed into cDNA through HiScript^®^ II Q RT SuperMix for qPCR (+gDNA Wiper) Kit. Finally, using SYBR Premix Ex Taq kit on ABI Q5 RT-PCR System (Applied Biosystems, Thermo Fisher Scientific, Inc., Waltham, MA, USA) for real-time quantitative PCR. Data were analyzed using ABI 7500 software version 2.03 (Applied Biosystems) using the 2^−ΔΔCT^ method. Specific primers for target mRNA and reference genes were designed as shown in Appendix A, and all of them were designed by the Pick Primers function in NCBI.

### 4.16. Cell Cycle Assay

A375, B16-F10 cells were seeded in 6-well plates and incubated with different concentrations of IR-817 (0, 0.31, 0.63, 1.25, 2.5 µM) for 24 h. After that, the cells were resuspended and washed with cooled PBS and fixed in 75% ethanol at 4 °C overnight. After PBS washing, PI was added to wrap the cells with a final concentration of 50 µg/mL for 30 min at 4 °C. Ultimately, using FACS flow cytometer (Novocyte, ACEA Bio, USA) to detect and data analysis using Novo Express.

### 4.17. Anti-Tumor Assays In Vivo

The zebrafish experiment was conducted using wild-type AB zebrafish. After the embryos were cultured to 48 hpf, GFP-A375 and CM-Dil-stained B16-F10 cells (Dil-B16-F10) suspension (200–400 cells/a fish) were injected into the space below the yolk sac of zebrafish by microinjector in a stereomicroscope (Nikon SMZ1000). We picked out the fish with sufficiently fluorescent melanoma cells and randomly divided them into diverse groups; each group contained 5 fish at least. Subsequently, adding IR-817 in the medium made the concentration 0, 0.31, and 1.25 µm. After 24 h, using a confocal laser-scanning microscope (FluoViewTM FV100 confocal microscope, Olympus Imaging) and a stereo microscope (Leica MZ FLII Stereo Fluorescence Microscope) was used to capture the image. All the experiments were approved by the Animal Research Ethics Committee of Southwest Medical University.

To study the anti-tumor effect in the mouse model, when the tumors were observable (tumor volume was around 100 mm^3^), the mice were randomly divided into two groups (Control, IR-817) with 5 mice in each group. The mice were intraperitoneally injected with 2 mg/kg IR-817 or commensurable PBS every two days. At the same time, recording their body weight and tumor volume. Tumor volume is calculated as follows: Volume (mm^3^) = A × B^2^ × 0.5, A (mm) and B (mm), respectively, represent the width and length of the tumor. Mice were sacrificed when tumor volume reached 1000 mm^3^ or ulceration occurred. Meanwhile, the organs and tumors of all mice were collected and weighed the tumors.

### 4.18. Biodistribution of IR-817 in Tissues

Firstly, the tumor and organ of mice were cut into frozen sections with a cryogenic microtome (Leica CM1950), fixed with ice-cold acetone for 15 min, washed with PBS, and stained with Hoechst33342 (2 μg/mL, 10 min) for nuclear staining. Finally, the samples were washed with PBS and observed under a fluorescence microscope (Leica DMi8).

### 4.19. Hematoxylin and Eosin (H&E) Staining

The collected viscera were fixed with 4% paraformaldehyde overnight, then dehydrated with ethanol, embedded in paraffin and cut into 3 μm-thick sections, and eventually stained with H&E, sealed with resin, and observed using a microscope (Nikon, NiE, Japan).

### 4.20. Immunohistochemistry

The tumor tissues of mice were embedded in paraffin after being fixed and dehydrated with 4% paraformaldehyde and ethanol. Then paraffin-embedded tissues were cut into 4 μm-thick sections in the slides, which were heated in antigenic repair solution at 95 °C for 20 min. After using the universal SP kit (mouse/rabbit streptavidin–biotin detection system) according to the manufacturer’s instructions, antibodies against CC-3, Ki-67, and E2f8 were individually incubated in tissue slides at 37 °C for 1 h. Subsequently, which incubated with HRP-conjugated goat anti-rabbit/mouse IgG for 15 min at 37 °C. In the end, horseradish peroxidase activity was detected using a DAB kit.

### 4.21. Statistical Analysis

The software used to analyze the data were ImageJ software (National Institutes of Health, Bethesda, MD, USA), Figdraw (www.figdraw.com, accessed data: 9 September 2022), GraphPad Prism 9 (GraphPad Software, San Diego, CA, USA), and SPSS 17.0 software (SPSS, Chicago, IL, USA). Statistical analyses were represented as mean ± SD. *p* < 0.05 were considered statistically significant.

## 5. Conclusions

In summary, we used a redox reaction of IR-808 and choline to produce the near-infrared fluorescent compound IR-817, which integrates diagnosis and treatment. In its diagnostic role, IR-817 showed good melanoma targeting imaging ability both in vitro and in vivo owing to its greater stability in FBS, which provided a basis for further application of IR-817 in vivo. At the same time, we also demonstrated that IR-817 entered cells through OATP transporters and targeted mitochondria. For the therapeutic effect, we identified that IR-817 induced cell apoptosis through the intrinsic mitochondrial apoptotic pathway and arrested cell cycle through the E2F/Cyclin/CDK pathway, and both features jointly endowed IR-817 with good anti-tumor ability. For further in vivo studies, IR-817 also showed remarkable antitumor ability in zebrafish and murine melanoma transplantation models. In conclusion, these results suggest that IR-817 is a promising integrated drug for melanoma diagnosis and treatment.

## Data Availability

Data is contained within the article and Appendix A.

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
