# Peer review of "Mitochondrion-Targeted NIR Therapeutic Agent Suppresses Melanoma by Inducing Apoptosis and Cell Cycle Arrest via E2F/Cyclin/CDK Pathway"

_pharmaceuticals, 2022, doi:10.3390/ph15121589_

Round 1

Reviewer 1 Report

At the outset, I would like to congratulate the authors on both the choice of subject matter and research concept which allow to demonstrate such a clear results. The manuscript is very interesting and is well-structured in general. The Authors conducted a number of assays (using both in vitro as well as in vivo experimental model) to reveal that IR-817 is a promising integrated drug for 623 melanoma diagnosis and treatment. In my opinion the topic of the study is original and relevant in the field of melanoma treatment and could be interesting for a reasonable number of scientists since malignant melanoma is the most fatal form of skin cancer worldwide, and earlier diagnosis and more effective therapies are required to improve prognosis. However there are some points that the Authors should address:

1.    MTT assay: the information concerning the preparation of IR-817 solutions should be added. Whether the DMSO was used and what was it final concentration(maximum) in the studied samples. It is very important to eliminate DMSO impact on cell viability.

2.    In vitro experimental panel – cell line selection – The Authors should pointed whether the studied melanoma cell lines were melanotic or amelanotic. It is very important because melanin biopolymers may impact on the diagnostic properties of the studied compound. Moreover, in my opinion this issue should be discussed.

In my opinion after these corrections the manuscript merit publication. 

Reviewer 2 Report

Journal- Pharmaceuticals

Title-"Mitochondrion-targeted NIR therapeutic agent suppresses melanoma by inducing apoptosis and cell cycle arrest via E2F/Cyclin/CDK pathway”

Reviewer comments

The topic of the present manuscript is vital and in demand of current research. However, the manuscript needs some changes and justification, which should be addressed to improve the quality of the paper. I recommend this manuscript be published in the "Pharmaceuticals" with Major Revision. The author needs to address the below comments/suggestions:

1.     In the introduction section, the authors have given some malignant melanoma statistics and given references (1 and 2), these are cross references. Instead, authors can follow the statistics of global agencies like WHO, the International Agency for Research on Cancer, and the American Cancer Society, Cancer Facts and Figures 2022, and use them as references.

2.     What was the stock concentration of MTT that was used in the MTT assay? Authors need to mention it.

3.     Authors need to provide C-13 data as well with the proton NMR for the structural elucidation of IR-817

4.     What are the passage numbers of A375, B16-F10 cell lines used for the mechanistic study of IR-817 ? Authors need to mention it in the section “Cell culture”, because, with the increased passage numbers, the phenotype and genotype of cell lines can change.

5.     Sometimes many therapeutic drugs showed excellent anticancer activity in-vitro and in-vivo but failed in the clinical trial because of their hemolytic toxicity.  Here authors also need to check the hemolytic toxicity on the red blood cell (erythrocytes) by in-vitro hemolysis assay and articles http://dx.doi.org/10.1016/j.biochi.2015.11.012 and doi:10.1371/journal.pone.0148877 can be used as reference.

6.     Authors need to give the NCBI gene number of various genes shown in Table S1 and which software was used to prepare primer sequences so that any researcher can repeat the data in the future.

7.     DAPI staining is a fundamental technique used for the detection of apoptosis and shrinkage, chromatin condensation, naked DNA, DNA fragmentation and loss of typical nuclear architecture are the most common apoptotic characteristics visualized with this staining. The results of DAPI staining which presented in Figure S9, is not proper; almost similar nuclear morphology appears in the untreated and treated group. Authors need to reperform this and label the apoptotic characteristics properly.

Round 2

Reviewer 2 Report

Journal- Pharmaceuticals

Manuscript- pharmaceuticals-2051462

Title- Mitochondrion-targeted NIR therapeutic agent suppresses melanoma by inducing apoptosis and cell cycle arrest via E2F/Cyclin/CDK pathway

Dear Editor,

The authors have fulfilled all the queries/comments as asked by reviewers previously. Hence, now the manuscript is well written. I believe that it is an excellent work for being published in the Pharmaceuticals. Finally, I recommend that the paper should be accepted for publication in the present form.

Decision- Accept